# Hidradenitis Suppurativa: Where We Are and Where We Are Going

**DOI:** 10.3390/cells10082094

**Published:** 2021-08-15

**Authors:** Emanuele Scala, Sara Cacciapuoti, Natalie Garzorz-Stark, Matteo Megna, Claudio Marasca, Peter Seiringer, Thomas Volz, Kilian Eyerich, Gabriella Fabbrocini

**Affiliations:** 1Division of Dermatology and Venereology, Department of Medicine Solna and Center for Molecular Medicine, Karolinska Institutet, 17176 Stockholm, Sweden; natalie.garzorzstark@ki.se (N.G.-S.); peter.seiringer@tum.de (P.S.); kilian.eyerich@ki.se (K.E.); 2Department of Clinical Medicine and Surgery, University of Naples Federico II, 80131 Naples, Italy; sara.cacciapuoti@libero.it (S.C.); mat24@libero.it (M.M.); claudio.marasca@gmail.com (C.M.); gafabbro@unina.it (G.F.); 3Department of Dermatology and Allergy, Technical University of Munich, 80802 Munich, Germany; thomas.volz@tum.de; 4Center for Allergy and Environment, Technical University, Helmholtz Center Munich, 80802 Munich, Germany; 5Unit of Dermatology, Department of Dermatology and Venereology, Karolinska University Hospital, 17176 Stockholm, Sweden

**Keywords:** hidradenitis suppurativa, diagnosis, pathogenesis, treatments, translational studies

## Abstract

Hidradenitis suppurativa (HS) is a chronic inflammatory skin disease primarily affecting apocrine gland-rich areas of the body. It is a multifactorial disease in which genetic and environmental factors play a key role. The primary defect in HS pathophysiology involves follicular occlusion of the folliculopilosebaceous unit, followed by follicular rupture and immune responses. Innate pro-inflammatory cytokines (e.g., IL-1β, and TNF-α); mediators of activated T helper (Th)1 and Th17 cells (e.g., IFN-γ, and IL-17); and effector mechanisms of neutrophilic granulocytes, macrophages, and plasma cells are involved. On the other hand, HS lesions contain anti-inflammatory mediators (e.g., IL-10) and show limited activity of Th22 cells. The inflammatory vicious circle finally results in pain, purulence, tissue destruction, and scarring. HS pathogenesis is still enigmatic, and a valid animal model for HS is currently not available. All these aspects represent a challenge for the development of therapeutic approaches, which are urgently needed for this debilitating disease. Available treatments are limited, mostly off-label, and surgical interventions are often required to achieve remission. In this paper, we provide an overview of the current knowledge surrounding HS, including the diagnosis, pathogenesis, treatments, and existing translational studies.

## 1. Introduction

Hidradenitis suppurativa (HS), also known as acne inversa, is a chronic inflammatory skin disease affecting ~1% of the global population [1]. HS typically occurs after puberty, with the average age of onset in the second or third decades of life and with a female predominance [2]. Approximately one third of patients are genetically predisposed [3]. Moreover, lifestyle factors, such as smoking and obesity, play a crucial role in the clinical course of HS [4,5,6].

Due to its chronic nature and frequently occurring relapses, HS has a great impact on the patients’ quality of life, deeply affecting social, working, and psychological aspects [2,7,8,9]. Therefore, early diagnosis is very important for HS patients in order to ensure the best possible course of this stigmatizing and painful disease. Here, we provide an overview of the current knowledge surrounding HS, including the diagnosis, pathogenesis, treatments, and translational studies.

## 2. Materials and Methods

This study is a narrative review. Pubmed and Scopus databases were reviewed up to June 2021. Search terms included “hidradenitis suppurativa”, “diagnosis”, “immunopathogenesis”, “cytokines”, “treatments”, “biological therapies”, “translational studies”, “in vivo animal models”, and “ex vivo human models” in the abstract and title. Duplicate articles, i.e., those with no full text available, were excluded. The obtained articles were analyzed by content analysis and compared further.

## 3. Diagnosis

### 3.1. Clinical Aspects

HS is usually localized in the apocrine gland-bearing areas (Figure 1). According to the modified Dessau definition, three diagnostic criteria should be matched: presence of typical lesions, typical locations, and chronicity [10]. Deep-seated painful nodules, abscesses, suppurative sinus tracts or tunnels, bridged scars, and double- and multi-ended comedones (also known as “tombstone comedones”) are described as typical lesions. Commonly, patients experience pain, malodor, a burning sensation, and an itch [11]. Chronicity, defined as two recurrences within the period of six months, is based on a complex pathogenesis underlying the disease; nodules and abscesses can rupture, bleed, and produce purulent discharge. The role of bacteria, infections, and superinfections in HS is still debated and controversial, but they probably have a role in promoting the chronicity of HS [12]. The persistence of the pathology entails the occurrence of dermal contracture and fibrosis of lesional skin. Intertriginous anatomic locations are typically involved, particularly the axillary, infra- and intermammary, inguinal, perineal, perianal, and gluteal regions [13]. HS may less frequently occur in the lower abdomen, suprapubic, retroauricular, nape, eyelids, and scalp [14,15]. Since diagnosis of HS may typically be delayed for years, in order to enhance a correct diagnosis, other factors should be investigated; above all, a family history of HS and history of other follicular occlusion diseases [16]. Moreover, the identification of pathogenic microbes should be evaluated and it could be useful in the therapeutic management, even if a primary infection is not the principal cause of HS [17]. Considering the lack of knowledge to date on the exact pathogenetic mechanism underlying the onset of HS, the phenotypic variants available today still receive criticisms and proposals for change. Three clinical subtypes of HS have been described: axillary-mammary, follicular, and gluteal [18]. According to another classification by van der Zee et al. [19], there are six phenotypes of HS: (i) regular type; (ii) frictional furuncle type; (iii) scarring folliculitis type; (iv) conglobata type; (v) syndromic type; and (vi) ectopic type. Recently, it was suggested to distinguish between a follicular subtype and an inflammatory phenotype, which is usually associated with a worse course of HS [20].

### 3.2. Severity Assessment

The first severity classification of HS was proposed by Hurley in 1989, categorizing patients into three stages based mainly on the presence of sinus tracts and scarring. The Hurley scoring system is a simple tool, nevertheless it is non-quantitative and static. Indeed, the use of this method in clinical studies may have significant limitations [21]. A more complex and detailed severity scoring method, called the Sartorius Scoring system, better suited to assess disease severity and grade of inflammation, was later proposed. Even though it is more accurate and precise in defining the severity of the disease, the Sartorius score may be time-consuming in routine clinical practice. Still today, the Hurley staging and the Sartorius scoring, together with HS Physician Global Assessment systems are the most commonly used assessment tools [22,23]. In addition, the HS Clinical Response (HiSCR), defined as the reduction in at least 50% of total abscess and inflammatory nodule count, with no increase in abscess count and no increase in draining fistula count relative to baseline, is used mainly to assess treatment effects in trials [24]. Recently, several other clinical measures for assessing HS disease severity were described, including the recently proposed HS Severity Score System (IHS4), the Acne Inversa Severity Index, and the Severity Assessment of HS score, which need further validation [25,26,27]. Ultrasound (US) aids in the diagnosis and assessment of HS disease severity [28]. Different subclinical lesions have been identified and categorized using US, and their recognition is able to modify a therapeutic approach [29]. Wortsman and colleagues proposed a sonographic scoring system for HS (SOS-HS) that combined the results of parameters included in Hurley’s classification with the relevant sonographic findings [28]. Finally, it is well known how HS may have a significant impairment of patients’ quality of life (QoL). Therefore, specific QoL tools have been developed and validated for HS: HiSQOL [30], HSQoL-24 [31], and patient-reported outcome (PRO) questionnaires [32]. Recently, a new graphical tool able to better describe HS burden, the so-called HIDRAdisk, has been introduced [33,34].

## 4. Etiology and Pathogenesis

### 4.1. Genetics

The question “To what extent is HS caused by genetic factors?” was placed in the top 10 most important uncertainties from a short list of 55 HS uncertainties in a priority setting partnership conducted for HS [35]. A positive family history was noted in approximately one-third of HS patients [3,5]. The understanding of the HS genetics begins in 1968 when Knaysi and colleagues [36] noted a family history in 3 out of 18 HS patients receiving surgical treatment. Fitzsimmons and colleagues [37,38] took up the story in 1984, publishing two case series and concluding that the pattern of inheritance suggested a single gene disorder inherited as an autosomal dominant trait. However, in the Fitzsimmons’ study, only 34% were affected, while for an autosomal dominant condition, the frequency of affected first-degree relatives should be 50%. The researchers argued that incomplete penetrance or incomplete case ascertainment may have contributed. In 2000, Von der Werth et al. [39] reviewed 14 of the surviving probands in a subsequent study, confirming autosomal dominant pattern of HS. In 2006, Gao et al. [40] identified a locus of interest on chromosome 1p21.1-1q25.3 in a four-generation Chinese family of HS patients. It was associated with a mutation in the γ*-Secretase* pathway and was later confirmed in multiple other families [41]. Further studies have identified 41 sequence variants in the γ*-Secretase* pathway, and 30 of them involve the nicastrin (*NCSTN)* gene which is critical to integrate and stabilize the different subunits of the γ*-Secretase* complex. However, evidence is growing that γ*-Secretase* mutations underpin only a minority of cases of HS, even in those individuals with a positive family history [41,42]. A Japanese study has also suggested that a pathogenic γ*-Secretase* mutation may be insufficient to produce the HS phenotype [43]. Moreover, any determination of the relative contribution of genetic in disease causation usually incorporates twin studies to compare disease risk in monozygotic and dizygotic pairs. In a recent cross-sectional study [44] on self-reported HS conducted from 2011 to 2016, data were collected from twins participating in the surveys of the nationwide Netherlands Twin Register. The narrow-sense heritability of HS was 77%, with the remainder of the variance due to unshared or unique environmental factors based on an age-adjusted model combining additive genetic factors and unshared or unique environmental factors, suggesting a stronger than previously assumed genetic basis of HS. Altogether, HS is inherited as a monogenic trait in a minority of patients. In the remaining cases, HS is seen as a complex, environmentally triggered disorder in a genetically predisposed host.

### 4.2. Lifestyle Factors

The association between HS and obesity has been well established, with numerous studies demonstrating greater disease severity with increasing body mass index (BMI) [45,46]. It has been speculated that obesity has a pathogenic effect on HS severity, due not only to increased skin fold friction, but also to low-grade systemic inflammation promoted by adipokine unbalance. Several adipocytokines produced by excess adiposity may serve to propagate the inflammatory cascades involved in HS. In fact, adipokine balance in HS is shifted towards increased expression of pro-inflammatory resistin and leptin, contributing to the augmentation of inflammatory processes in the skin of patients with HS [47].

HS patients are at increased risk of developing other disorders, such as high cholesterol, high blood pressure, and diabetes [46,48]. In particular, a higher prevalence of diabetes mellitus (DM) among patients with HS compared to control subjects was observed [49], with prevalences ranging from 7.1% to 24.8%. In 2 meta-analyses of 12 and 7 studies, the pooled odds of DM among patients with HS were 2.17 (95% CI, 1.9–2.6) [50] and 2.8 (95% CI, 1.8–4.3) [51] times that of the control individuals, respectively. Clinicians may consider incorporating weight loss and the elimination of potential dietary triggers [52], as adjunct interventions, into the management plan. In particular, insulin, insulin-like growth factor 1 (IGF-1), and branched chain amino acids (including leucine, isoleucine, as well as valine) are found to be increased in red meat and dairy products. Overall, they lead to the activation of mammalian target of rapamycin (mTOR), whose expression was found to be increased in HS lesions compared to non-HS lesions in a previous study [53]. Interestingly, mTOR signaling has been implicated in promoting adipogenesis and lipogenesis, as well as the accumulation of triacylglycerols in sebaceous glands [53,54]. Among the environmental factors, it is worthy to mention the potential role of non-steroidal anti-inflammatory drugs (NSAIDs). They are commonly used by HS patients to treat pain and inflammation [55]. However, NSAIDs decrease immunity and have been associated with infection onset and/or worse outcomes [56,57]. Stopping NSAIDs might be an important environmental measure to take for patients’ improvement.

Cigarette smoking is another trigger factor associated with HS [58]. Epidemiological studies have shown that ~90% of HS patients are smokers [22]. The exact mechanisms by which smoking contributes to HS pathogenesis remain unclear. However, nicotine has been related to pathogenic events in HS such as epidermal hyperplasia, follicular plugging, neutrophil chemotaxis, cytokine production by keratinocytes, and down regulation of antimicrobial peptides (AMPs) [59]. Furthermore, it has been shown that cigarette smoking may be detrimental to the healing of HS lesions; it reduces cutaneous blood flow, and tissue oxygenation, and impairs wound healing. Thus, smoking cessation should be included as part of the therapeutic management of HS patients [60,61].

Finally, the factor “stress” is also associated with HS. Its contribution still remains to clarify in HS. The latter has classically been associated with apocrine glands which respond to circulating adrenergic mediators during stress conditions [62]. However, the pathophysiology involves follicular occlusion rather than an apocrine disorder, as previously thought.

### 4.3. Pathogenic Events

The primary defect in HS pathophysiology includes the occlusion and the consequent inflammation of the hair follicle; these conditions, together with both innate and adaptive immune dysregulation, are essential for the development of HS [2]. Indeed, 95% of early HS lesions present hyperkeratosis, occlusion of the follicular unit and an associated perifolliculitis [4]. In this scenario, bacterial colonization is considered a secondary pathogenic factor that can worsen HS [2,4]. Indeed, HS lesions formation starts with follicular occlusion which leads to dilatation followed by rupture, resulting in extrusion of the follicular contents, including keratin and bacteria, into the surrounding dermis with the induction of an intense inflammatory response from neutrophils and lymphocytes (Figure 2). The perifollicular inflammatory cellular infiltrate causes abscess formation, leading to the destruction of the pilosebaceous unit [2,4,63,64].

The primary cause of follicular occlusion has not been identified. Melnik and Plewig [65] recently proposed the concept of HS as an auto-inflammatory disease characterized by dysregulation of the gamma-secretase/Notch pathway which has a key role in maintaining the inner and outer root sheath of the hair follicle and skin appendages. Even if it is clear that HS is a follicular disease and not a primary infectious disease, the role of bacteria seems to be very important in HS pathophysiology. Indeed, releasing bacteria within the dermis after follicular rupture represent a trigger for local inflammatory response. Particularly, *Staphylococcus lugdunensis* seems to dominate in HS nodules and abscesses, followed by a polymicrobial anaerobic microflora mainly comprising prevotella and porphyromonas, but also milleri group streptococci, and actinomycetes [66,67]. Bacterial dysbiosis is present at the skin’s surface, in the follicle, and at distinct body sites in HS compared with normal skin with the decrease in the relative abundance of skin commensals, and the increase of opportunistic anaerobic pathogens being found in HS with respect to healthy skin [68,69]. Bacteria may be difficult to eradicate from HS lesions since they form biofilms (mainly found in in sinus tracts or in the infundibulum), further sustaining chronic inflammation which guides HS [70]. In addition, a dysregulation of tryptophan catabolism (with overactivation of the kynurenine pathway and consequent stimulation of cytokines by inflammatory cell infiltrates) and of an aryl hydrocarbon receptor (AHR) (with impaired production of bacteria-derived AHR agonists and decreased incidence of AHR ligand-producing bacteria) has been found in HS skin, thus providing a mechanism linking the immunological and microbiological features of HS lesions as the case of tryptophan metabolism and microbiota cross talk reported in Chron’s disease [71,72].

### 4.4. Main Cytokine Networks in HS

Immune cells and keratinocyte-mediated products are widely accepted as key players in HS pathogenesis [73]. Indeed, both pro-inflammatory (e.g., IL-1β, TNF-α, IL-23, and IL-17) and anti-inflammatory cytokines (e.g., IL-10) are found to be increased in HS lesional and perilesional skin [74,75,76,77]. It was proposed that the release of follicular content activates the NLRP3 inflammasome, and caspase-1 leading to IL-1β production into the skin [75,77]. Indeed, elevated levels of caspase-1 with enhanced mRNA expression of NRLP3 as well as IL-1β were detected in HS lesions [75,77,78].

IL-1β induces the production of matrix metalloproteinases (MMPs) (may be involved in the early rupture of the hair follicle units and the later loosening of epidermal cell–cell junctions during tunnel formation) and selected chemokines which recruit neutrophils at the skin level [79]. The recruitment of these cells contribute to the inflammatory cytokine production and pus formation in HS skin.

On the other hand, keratin fibers and other debris produced (as result of tissue destruction and scarring) can be recognized by macrophage and dendritic cells (DCs) through toll-like receptors (TLRs) [4,5]. All of these events produce an increased amount of pro-inflammatory cytokines including TNF-α which positively correlates with disease severity [80].

TNF-α induces a wide range of immune-cell-attracting chemokines and contributes to endothelial activation, favoring immune cell infiltration [73]. Therefore, HS lesions are characterized by granulocytes, T cells, B lymphocytes, plasma cells, and monocytes (which differentiate into macrophages and DCs) [1,73]. Additionally, TNF-α acts as an upstream mediator of T cell differentiation into T helper (Th)1, and Th17 cells which produce interferon (IFN)-γ and IL-17, respectively [1,79].

The production of IFN-γ and IL-17 is further supported by mTOR complex signaling, whose relevance might be deduced from the reported increased mTOR expression in HS lesions [81,82,83,84,85]. As previously reported by Schroder et al., IFN-γ induces Th1-attracting chemokines (e.g., CXCL10) and activates dermal endothelial cells, allowing the immune cell infiltration from the bloodstream [86].

Elevated levels of IL-17 have been observed in HS lesions [87,88]. Coherently, high levels of IL-23 (which drives IL-17 production) have been also found in HS lesional skin [74,88]. Interestingly, IL-17 might drive the production of IL-1β by keratinocytes via the activation of NLRP3 inflammasome, and caspase-1 [5]. Moreover, IL-36 cytokines, which belong to the IL-1 family, are able to mediate Th1 and Th17 responses [89,90]. Together, these findings suggest a positive feedback loop between IL-1 family members and IL-17.

Among Th17 cytokines, IL-26 has also been found to increase in the plasma and skin of HS patients [88]. It leads to the production of pro-inflammatory cytokines through its receptor (IL-10R2 and IL-20R1) but it can also bind to cell-free DNA and subsequently activate TLRs pathways [91]. Furthermore, IL-26 exerts direct antimicrobial activities having the ability to open pores in bacterial membrane [5,91]. Interestingly, IL-26 bactericidal activity might be ineffective in HS patients. Indeed, IL-26-related antimicrobial, cytotoxic, and phagocytic activities were found to be lower in HS patients compared to healthy donors [88].

While Th1 and Th17 cells and their main mediators become abundant in HS skin, Th22 cells and IL-22 are not [79]. HS lesions are characterized by a reduced infiltration of IL-22-secreting cells and impaired production of IL-22 by these cells [92,93]. It is worthy to note that IL-22 production is enhanced by Notch signaling, which is defective in some HS patients [92,94]. IL-22 deficiency has also been linked to increased IL-10 production, which might be induced, among others by IL-1β [87]. The inflammatory vicious circle finally results in pain, purulence, tissue destruction, and scarring (Figure 3).

## 5. Treatments

### 5.1. From Conventional Therapies to Molecular Treatments

Current therapeutic concepts of HS are mainly based on three pillars: (1) patient education; (2) medical treatment of inflammation; and (3) surgical therapy of fistulas, nodules, and scar tissue [95] (Table 1). Depending on clinical presentation and HS severity, a combination of all pillars is often necessary in order to maintain disease control. Independent of the duration and severity of HS, all patients should follow basic therapeutic measures, including self-education about the disease, pain management, maintaining high hygiene standards, cessation of smoking, and losing weight, amongst others [1,96].

Topical antibiotics and antiseptics are frequently used, although there is little evidence for efficacy [97,98]. Systemic antibiotics (e.g., tetracyclines, clindamycin, rifampicin) are considered first-line therapy according to guidelines and expert opinions, although HS is not primarily an infectious disease, and none of them are approved by the Federal Drug Agency (FDA) or the European Medicines Agency (EMA) for the treatment of HS [1,63,95,99]. Their mode of action is believed to be immunomodulatory by reducing NFκB activation, resulting in diminished production of pro-inflammatory cytokines [12]. Combined with surgery, optimized antibiotic treatments may be promising in severe HS [100]. Ertapenem, an intravenous broad spectrum antibiotic, is highly effective and can be used as an adjunctive treatment to surgery [100,101,102]. However, it is a β-lactamic carbapenem with no anti-inflammatory proprieties and is reserved as third-line therapy for a single six-week course as rescue therapy or during surgical planning [98,101]. No unrestricted recommendation can be given for the use of conventional immunosuppressant or immunomodulatory drugs including prednisolone [103], dapsone [104], retinoids [105], and ciclosporin [106], amongst others, as there is little or conflicting evidence for their efficacy [63,98]. Directly targeting cytokines by biologics represents the most promising anti-inflammatory treatment currently available in HS. Adalimumab (ADA), an anti-TNF-α-antibody, is the only EMA- and FDA-approved therapy for the moderation of to severe HS at the moment [107]. As numerous clinical trials evaluating biologics and other modern immunotherapies are being conducted, it should only be a matter of time until the therapeutic landscape of HS is expanded and patient-centered precision medicine becomes reality [1,108]. Despite these already achieved and coming-up advances, structural alterations of HS-affected skin and subcutaneous tissue presenting with fistulas, sinus tracts, or large scarred areas should primarily be targeted by radical surgical excision [63,109].

**Table 1 cells-10-02094-t001:** Therapeutic approach of HS.

Hurley Staging (HS Severity)	Clinical Features	Recommended Treatments	Surgery
Stage I (mild)	Single or multiple abscesses without sinus tracts and scarring	Patient education: maintaining hygiene standards, cessation of smoking, losing weight * Topical clindamycin, antisepticsIntralesional corticosteroids	Incision and drainage Deroofing Laser
Stage II (moderate)	Recurrent abscesses with sinus tracts and scarring	Systemic antibiotics: tetracycline, clindamycin + rifampicin, rifampicin + moxifloxacin + metronidazole, ertapenem Immune modulators: prednisolone, retinoids, dapsone, cyclosporine	Local intervention Deroofing Laser
Stage III (severe)	Diffuse or multiple interconnected sinus tracts and abscesses across the entire area	Biologics: adalimumab, bimekizumab (phase III)/secukinumab (phase III), others (Table 2)	Extensive radical surgical excision

* Patient education is essential in all stages. Data from multiple sources [97,99,100,103,104,105,106,107,110,111,112,113,114,115,116,117,118]. HS, hidradenitis suppurativa.

### 5.2. Predictive Markers of Therapeutic Response

The identification of predictive biomarkers of therapeutic response is one of the greatest challenges of modern medicine, particularly in a pathology as HS with a high percentage of therapeutic failures. Although several studies have shown that anti-TNF-α agents [e.g., infliximab (IFX), and ADA] clinically reduce the disease activity in HS [119], the efficacy of these agents is inconstant in some cases. Identifying predictive markers of response is, thus, of strong interest. In a prospective pilot study, Montaudié et al. [120] aimed to determine if the baseline inflammatory profile of HS patients could predict the response to treatment with IFX, demonstrating that initial levels of high-sensitivity C-reactive protein (CRP) and IL-6 are potential response markers for IFX treatment in HS, with a greater reliability for IL-6, considering the multiple confounding factors when using CRP levels. More recently, Cao et al. [121] focused on potential predictive biomarkers of ADA response in HS patients, evaluating baseline and week-12 plasma samples from the PIONEER studies [122], to assess the levels of circulating proteins by multiplex and enzyme-linked immunosorbent assays. The authors suggest that chemokine (C-C motif) ligand (CCL) 16 (HCC-4), calprotectin, and fractalkine could be potential predictive biomarkers of ADA response in HS. Blok et al. [123] hypothesized the use of leukotriene A4-hydrolase (LTA4H) levels as predictive biomarkers of a therapeutic response in HS patients treated with ustekinumab (UST), concluding that low LTA4H concentrations with mild disease severity may be predictive of the effectiveness of UST. Other authors [124] focused on the analysis of messenger RNA (mRNA) expression levels of six selected inflammation-related miRNAs in lesional and perilesional skin samples of HS patients and in healthy controls, observing a significant overexpression of miRNA-155-5p, miRNA-223-5p, miRNA-31-5p, miRNA-21-5p, and miRNA-146a-5p in lesional HS skin compared to healthy controls. These miRNAs have been proposed as potential disease biomarkers, suggesting their therapeutical manipulation to target the inflammatory pathway in HS. Finally, a very innovative approach was recently proposed, based on salivary “liquid biopsy” associated to vibrational spectroscopy to develop a personalized medical approach for HS patients’ management [125].

### 5.3. Future Treatments and Personalized Therapies

Advances in understanding HS pathogenesis are leading to the future expansion of the disease treatment armamentarium [126]. Indeed, several small molecules and biologics are under investigation, for moderate-to-severe HS therapy (Table 2). With phase III trials ongoing, the anti-IL-17 agents bimekizumab (which blocks IL-17 A and F) and secukinumab (which blocks IL17A) are in the most advanced stage of clinical development showing promising results, based on the extensive evidence of the activation and upregulation of the IL-17 pathway in HS inflammation [117]. Even if controversial data regarding possible paradoxical reactions have been described [127], recent studies have showed only partially confirmed effectiveness of secukinumab in the treatment of moderate-severe HS, since HiSCR was achieved only by 41% of patients at week 28 [128]. Bermekimab, which targets IL-1α, has shown resolution of inflammatory lesions and pain in a phase II trials on moderate-to-severe HS while phase II trials are also being conducted for spesolimab which acts on neutrophil recruitment and activation by targeting IL-36 [117]. In addition, modulation of several proinflammatory cytokines and chemokines, such as TNF-α, IL-12/IL-23, CXCL9, and CXCL10, through administration of the oral phosphodiesterase-4 inhibitor apremilast was positively tested in mild to moderate HS in a phase II study [129]. However, drugs which are being tested for HS are very numerous and further include also complement 5a (C5a) and C5a receptor blocking agents (avacopan and IFX-1), janus kinase (JAK), signalling inhibitors (tofacitnib and upadacitinib), anti-CD40 (iscalimab), and anti-iIL23 agents (guselkumab and risankizumab) [117,130]. Furthermore, case reports or case series also show good results for other drugs, such as certolizumab pegol (anti-TNF-α) [131,132,133], tildrakizumab (anti-IL23) [134], ixekizumab (anti-IL17) [135], and brodalumab (anti-IL17 receptor) [136]. In this crowded future treatment scenario, it should be emphasized that HS is a complicated disease that involves both pathological and environmental factors. Hence, treating HS will continue to involve a multidisciplinary approach with monotherapy, often tending not to be efficacious or enough alone. Future treatment should be guided by personalized therapy, biomarkers, and pharmacogenomics (precision medicine), which should drive the therapeutic choice in order to maximize clinical outcomes and limit adverse events following patients peculiarities at genetic and phenotypic level [137].

**Table 2 cells-10-02094-t002:** Ongoing and upcoming interventional trials on HS.

Intervention/Compound	Target	Type	Phase	Number of Studies	ClinicalTrials.gov Identifier
Avacopan	C5a receptor	systemic drug	2	1	NCT03852472
Bimekizumab	IL-17A, IL-17F	systemic drug	3	2	NCT04242446, NCT04242498
Iscalimab (CFZ533), LYS006	CD40, LTA4H (respectively)	systemic drug	2	1	NCT03827798
CSL324	G-CSF receptor	systemic drug	1	1	NCT03972280
Imsidolimab	IL-36 receptor	systemic drug	2	1	NCT04856930
INCB05470	JAK-1	systemic drug	2	1	NCT04476043
KT-474	IRAK4	systemic drug	1	1	NCT04772885
LY3041658	ELR ^+^ CXC chemokines	systemic drug	2	1	NCT04493502
Metformin	unknown (anti-inflammatory)	systemic drug	3	1	NCT04649502
PF-06650833, PF-06700841, PF-06826647	IRAK4, TYK2+JAK1, TYK2 (respectively)	systemic drug	2	1	NCT04092452
Risankizumab	IL-23p19	systemic drug	2	1	NCT03926169
Secukinumab	IL-17A	systemic drug	3	3	NCT04179175, NCT03713619, NCT03713632
Spesolimab	IL-36 receptor	systemic drug	2	2	NCT04876391, NCT04762277
Tofacitinib	pan-JAK	systemic drug	2	1	NCT04246372
Upadacitinib	JAK-1	systemic drug	2	1	NCT04430855
Local therapy (wound dressings, creams, gels, sclerotherapy, AMP, triamcinolone)		local therapy	all	9	NCT04648631, NCT04194541, NCT04388163, NCT04354012, NCT04541550, NCT02805595, NCT04582669, NCT04756336, NCT04414514
Laser treatment		Laser	n.a.	2	NCT04508374, NCT03054155
Surgery (different procedures +/−ADA or NPWT+/−i)		Surgery	n.a.	4	NCT04526561, NCT04325607, NCT03784313, NCT03221621
Other (acupuncture, HS management, electronic reporting)		Other	n.a./4	3	NCT04218422, NCT04200690, NCT04132388

ADA, adalimumab; AMP, Allogeneic Micronized Amniotic Membrane Product; CD, cluster of differentiation; G-CSF, granulocyte colony-stimulating factor; IL, interleukin; IRAK4, interleukin-1 receptor associated kinase 4; JAK, janus kinase; LTA4H, leukotriene A4 hydrolase; n.a., not applicable; NPWT+/−i, negative pressure wound therapy with(+)/without(−) instillation; TNF, tumor necrosis factor.

## 6. Perspectives of Translational Studies in HS

### 6.1. In Vivo Animal Models

Adequate models reflecting hallmarks of HS pathogenesis are a prerequisite to not only better characterize the molecular events underlying initiation and progression of this disease, but also to allow the discovery and verification of novel therapeutic targets. The first murine HS models were engineered based on the finding that γ-secretase-associated polymorphisms were found in families with an autosomal dominant HS-like syndrome [138]. Although some features, such as infundibular plugging and cyst formation, occurred in these transgenic γ-secretase mouse models, clinical characteristics of HS, such as inflammation, abscess formation, fistulas, and scarring, were absent [139]. With *NCSTN* being the most frequently mutated gene in familial HS and also an integral component of the multimeric gamma-secretase complex, Yang et al. [140] recently presented a keratin 5-cre-driven epidermis-specific *NCSTN* conditional knockout mouse. Interestingly, this mutant showed key features of HS, including hyperkeratosis of hair follicles with keratotic plug formation, and inflammation of the hair follicles. Although some features of HS, such as abscesses, could not be recapitulated, and further studies will be needed to further validate the *NCSTN*^flox/flox^;K5-Cre mice, they might be the most promising ones amongst the transgenic murine lines, currently. Major issues in advancing with murine models are possibly not least related to the fact that mice and humans differ in terms of dermal thickness, hair follicle cycles, and hair distribution. Therefore, xenograft models could be particularly interesting for HS research, as they could overcome these obstacles and open up new perspectives for a better understanding of complex inflammatory skin diseases as they did for psoriasis, and alopecia areata [141,142]. While for other inflammatory skin diseases, such as AD and psoriasis, human tissue is limited for xenografts in terms of size and access, substantial amounts of HS tissue could be retrieved from surgical interventions. This potential was recognized, and recently, a first HS xenograft mouse model was published by Quartey et al. [143] using human HS lesions grafted onto immunocompromised mice. However, the presented HS xenograft model exhibited altered histopathological features with dyspigmentation, and will need further refinement, such as local and systemic transfer of human HS immune cells to reconstitute the immunodeficient mice.

### 6.2. Ex Vivo Human Models

With a long way still ahead to convincing murine HS models, promising ex vivo models using lesional human HS skin have been proposed that mimic the in vivo situation by maintaining the patients’ skin architecture and allowing investigation of drugs [144]. Vossen et al. [145] developed a punch biopsy transwell culture and revealed that, amongst a variety of anti-inflammatory drugs, including modern biologicals, TNF-α inhibitors and prednisolone showed the highest inhibitory effect on proinflammatory cytokines and AMPs in HS lesional skin which fits to the clinical experience regarding TNF-inhibitors. Not only for drug screen purposes but also in terms of gaining novel insights into HS pathogenesis, these models are beneficial. In this context, Vossen et al. [146] were able to analyze the contribution of IL-1 for driving inflammation in HS. However, explant models still have limitations in terms of defining the optimal balance between adequate nutrient diffusion and tissue volume for recapitulation of most of the in vivo epidermal-stromal interactions [144]. The currently considered optimal model for ex vivo studies for HS is the 3D-SeboSkin model in which explant skin is co-cultured in direct contact with SZ95 sebocytes [147]. This model shows normal skin thickness and architecture, whereas explant models without sebocytes in direct contact exhibit epidermal degeneration and basal keratinocyte vacuolization. Furthermore, lipid accumulation is more abundant in the 3D-SeboSkin model than in the conventional explant model. Until research may become independent of patient biopsies by using human-like organs printed on chips and controlled by microfluidic systems [148], current cell culture and explant approaches will be further improved and tailored to these research questions.

## 7. Conclusions

HS is a chronic inflammatory skin disorder with many contributing factors. It can greatly impact patients’ quality of life and social and work activities due to frequent disease relapses with painful and foul-smelling lesions. Therefore, prompt treatment is necessary to reduce the HS burden. Available medicaments are limited, mostly off-label, and surgical interventions are often required to achieve remission. Immunomodulatory therapies targeting specific cytokines might represent an effective option in controlling this condition. Adalimumab, a TNF-α blocker, is the only FDA-approved biological drug for the moderation of severe HS, whereas the majority of cytokines’ selective inhibitors (e.g., IL-12/23 and IL-17) are still in their early phases. However, clinical efficacy of all novel treatments reported so far is modest. Thus, HS remains a challenging disease to treat, and patients often need a multidisciplinary approach. Future translational studies are needed to better define this enigmatic skin disorder.

## Figures and Tables

**Figure 1 cells-10-02094-f001:**
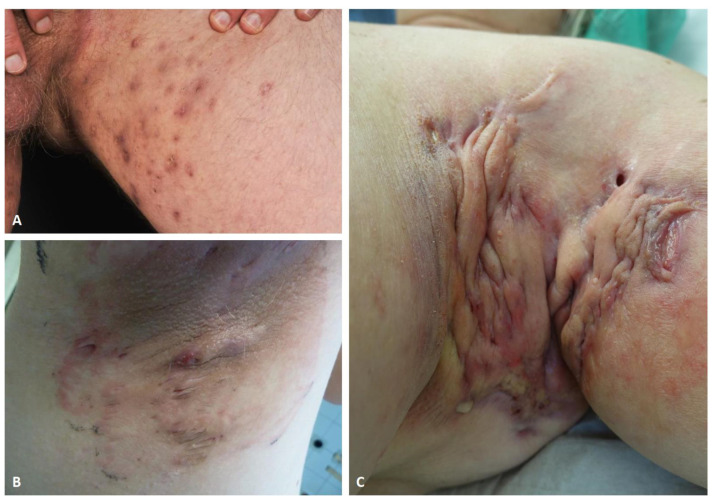
Clinical images of typical HS lesions in groin (**A**), and axilla (**B**,**C**). Superficial papules, small abscesses without scarring or sinus tracts, Hurley stage I (**A**). Multiple, recurrent abscesses with initial sinus tracts and cicatrization, Hurley stage II (**B**). Diffuse involvement of the axillary region with large abscesses, interconnected tracts, and scarring, Hurley stage III (**C**).

**Figure 2 cells-10-02094-f002:**
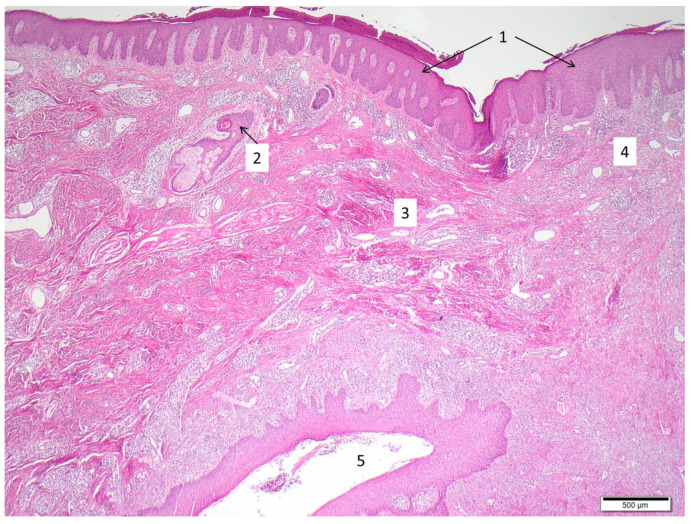
Typical histological features of HS. Sample acquired by punch biopsy from gluteal region. Hyperparakeratosis and papillomatosis (1), follicular hyperkeratosis and perifolliculitis (2), fibrosis (3), abscess-like accumulation of neutrophils and spotted infiltrate of lymphocytes/plasma cells (4), epithelialized sinus tract with surrounding inflammatory reaction (5).

**Figure 3 cells-10-02094-f003:**
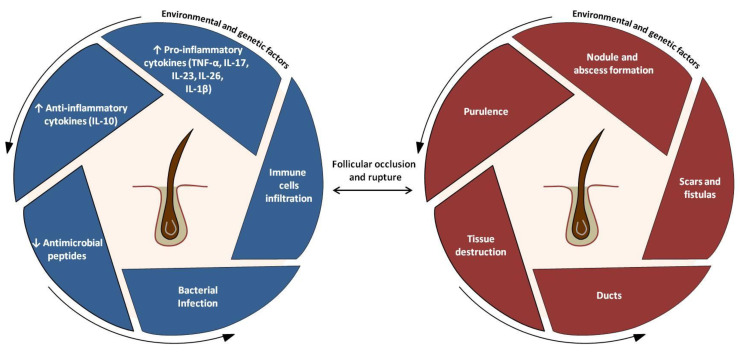
Pathophysiology of HS: a schematic overview.

## Data Availability

The Figure 3 “Pathophysiology of HS: a schematic overview” has been adapted from the following source: https://www.humiradermpro.com/mechanism-of-action/hs (accessed on 29 June 2021).

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
