# Peer review of "Hidradenitis Suppurativa: Where We Are and Where We Are Going"

_cells, 2021, doi:10.3390/cells10082094_

Round 1
Reviewer 1 Report
Well documented and well written paper on an important topic, Hidradenitis Suppurative.
I would ask the authors to expand the information given on Quality of Live (top lines of page 4). Recently, various disease specific QoL questionnaires have been developed and validated, and should be mentioned, in the same way you have mentioned the HIDRAdisk. These are:
Kirby, J.; Thorlacius, L.; Villumsen, B.; Ingram, J.; Garg, A.; Christensen, K.; Butt, M.; Esmann, S.; Tan, J.; Jemec, G. The
Hidradenitis Suppurativa Quality of Life (HiSQOL) score: Development and validation of a measure for clinical trials. Br. J.
Dermatol. 2019, 183, 340–348. [CrossRef]
Marrón, S.; Gómez-Barrera, M.; Aragonés, L.T.; Díaz, R.D.; Rull, E.V.; Álvarez, M.M.; Puig, L. Development and Preliminary
Validation of the HSQoL-24 Tool to Assess Quality of Life in Patients with Hidradenitis Suppurativa. Actas Dermo-Sifiliogr. 2019,
110, 554–560. [CrossRef] [PubMed]
Chiricozzi, A.; Bettoli, V.; De Pità, O.; Dini, V.; Fabbrocini, G.; Monfrecola, G.; Musumeci, M.; Parodi, A.; Sampogna, F.; Pennella,
A.; et al. HIDRAdisk: An innovative visual tool to assess the burden of hidradenitis suppurativa. J. Eur. Acad. Dermatol. Venereol.
2018, 33, e24–e26. [CrossRef]
Kimball, A.B.; Sundaram, M.; Banderas, B.; Foley, C.; Shields, A.L. Development and initial psychometric evaluation of patientreported
outcome questionnaires to evaluate the symptoms and impact of hidradenitis suppurativa. J. Dermatol. Treat. 2018, 29,
152–164. [CrossRef] [PubMed]
Finally, on page 9, when you talk about treatments, you have not mentioned Certolizumab pegol. Below are some references:
- Holm JG, Jørgensen AR, Yao Y, Thomsen SF. Certolizumab pegol for hidradenitis suppurativa: case report and literature review. Dermatol Ther. 2020 Nov 2:e14494. doi: 10.1111/dth.14494.
- Wohlmuth-Wieser I, Alhusayen R. Treatment of hidradenitis suppurativa with certolizumab pegol during pregnancy. Int J Dermatol. 2020 Nov 28. doi: 10.1111/ijd.15286. Epub ahead of print.PMID: 33247833.
- Esme P, Akoglu G, Caliskan E: Rapid Response to Certolizumab Pegol in Hidradenitis Suppurativa: A Case Report.Skin Appendage Disord 2020. doi: 10.1159/000511284
Author Response
Point to point reply:
We are very thankful to the Reviewer for the constructive suggestions, and we hope that the revised version of our manuscript can now reach high enough quality to be published in Cells.
Comment #1: Well documented and well written paper on an important topic, Hidradenitis Suppurative. I would ask the authors to expand the information given on Quality of Live (top lines of page 4). Recently, various disease specific QoL questionnaires have been developed and validated, and should be mentioned, in the same way you have mentioned the HIDRAdisk. These are: Kirby, J.; Thorlacius, L.; Villumsen, B.; Ingram, J.; Garg, A.; Christensen, K.; Butt, M.; Esmann, S.; Tan, J.; Jemec, G. The Hidradenitis Suppurativa Quality of Life (HiSQOL) score: Development and validation of a measure for clinical trials. Br. J. Dermatol. 2019, 183, 340–348. [CrossRef]; Marrón, S.; Gómez-Barrera, M.; Aragonés, L.T.; Díaz, R.D.; Rull, E.V.; Álvarez, M.M.; Puig, L. Development and Preliminary Validation of the HSQoL-24 Tool to Assess Quality of Life in Patients with Hidradenitis Suppurativa. Actas Dermo-Sifiliogr. 2019,110, 554–560. [CrossRef] [PubMed]; Chiricozzi, A.; Bettoli, V.; De Pità, O.; Dini, V.; Fabbrocini, G.; Monfrecola, G.; Musumeci, M.; Parodi, A.; Sampogna, F.; Pennella,A.; et al. HIDRAdisk: An innovative visual tool to assess the burden of hidradenitis suppurativa. J. Eur. Acad. Dermatol. Venereol.2018, 33, e24–e26. [CrossRef]; Kimball, A.B.; Sundaram, M.; Banderas, B.; Foley, C.; Shields, A.L. Development and initial psychometric evaluation of patientreported outcome questionnaires to evaluate the symptoms and impact of hidradenitis suppurativa. J. Dermatol. Treat. 2018, 29,152–164. [CrossRef] [PubMed] Reply: Thanks a lot for your comment, all the suggested references have been now cited.
Comment #2: Finally, on page 9, when you talk about treatments, you have not mentioned Certolizumab pegol. Below are some references: Holm JG, Jørgensen AR, Yao Y, Thomsen SF. Certolizumab pegol for hidradenitis suppurativa: case report and literature review. Dermatol Ther. 2020 Nov 2:e14494. doi: 10.1111/dth.14494; Wohlmuth-Wieser I, Alhusayen R. Treatment of hidradenitis suppurativa with certolizumab pegol during pregnancy. Int J Dermatol. 2020 Nov 28. doi: 10.1111/ijd.15286. Epub ahead of print.PMID: 33247833; Esme P, Akoglu G, Caliskan E: Rapid Response to Certolizumab Pegol in Hidradenitis Suppurativa: A Case Report.Skin Appendage Disord 2020. doi: 10.1159/000511284. Reply: Thank you for pointing this out. Certolizumab and the suggested references have been added to the manuscript (paragraph 5.3).
Reviewer 2 Report
This is a well-written review on HS field. A few precisions on microbiology, pathogenesis, triggering and environmental factors could improve this article.
About microbiology :P5, line 204-208: When prolonged cultures (1 week) and/or bacterial metagenomics are performed, a rich anaerobes flora (including mostly porphyromonas and prevotella), but also actinomycetes and S milleri are isolated from HS lesions, varying with Hurley severity and, contrarily to what the authors wrote, there is very little S aureus in HS lesions, but instead rather S lugdunensis. The chosen references are not appropriate: the 4 main teams who extensively worked on microbiology should be cited:
- Guet-Revillet H, Coignard-Biehler H, Jais JP, Quesne G, Frapy E, Poirée S, Le Guern AS, Le Flèche-Matéos A, Hovnanian A, Consigny PH, Lortholary O, Nassif X, Nassif A, Join-Lambert O. Bacterial pathogens associated with hidradenitis suppurativa, France. Emerg Infect Dis. 2014 Dec;20(12):1990-8. doi: 10.3201/eid2012.140064.
- Guet-Revillet H, Jais JP, Ungeheuer MN, Coignard-Biehler H, Duchatelet S, Delage M, Lam T, Hovnanian A, Lortholary O, Nassif X, Nassif A, Join-Lambert O. The Microbiological Landscape of Anaerobic Infections in Hidradenitis Suppurativa: A Prospective Metagenomic Study. Clin Infect Dis. 2017 Jul 15;65(2):282-291. doi: 10.1093/cid/cix285.
- Ring HC, Bay L, Nilsson M, Kallenbach K, Miller IM, Saunte DM, Bjarnsholt T, Tolker-Nielsen T, Jemec GB. Bacterial biofilm in chronic lesions of hidradenitis suppurativa. Br J Dermatol. 2017 Apr;176(4):993-1000. doi: 10.1111/bjd.15007.
- Naik HB, Jo JH, Paul M, Kong HH. Skin Microbiota Perturbations Are Distinct and Disease Severity-Dependent in Hidradenitis Suppurativa. J Invest Dermatol. 2020 Apr;140(4):922-925.e3. doi: 10.1016/j.jid.2019.08.445.
- Schneider AM, Cook LC, Zhan X, Banerjee K, Cong Z, Imamura-Kawasawa Y, Gettle SL, Longenecker AL, Kirby JS, Nelson AM.J Loss of Skin Microbial Diversity and Alteration of Bacterial Metabolic Function in Hidradenitis Suppurativa. Invest Dermatol. 2020 Mar;140(3):716-720. doi: 10.1016/j.jid.2019.06.151.
P7, l 279: the cited ref 77 does not support the assumption of “low efficacy of AB”. In this reference, the authors state : “Ertapenem is highly effective but is reserved as third-line therapy for a single 6-week course as rescue therapy or during surgical planning.” Ertapenem happens to be an antibiotic strategy targeted according to Hurley 3 microbiology, demonstrating the correlation between efficacy and targeted antibiotherapy, instead of random antibiotics which do have a “low efficacy”.
About pathogenesis, authors could also mention AHR pathway (Prens et al) involvement as well as Tryptophane pathway (Guenin-Macé et al). Indeed, these 2 pathways interact and tryptophane pathway has also been involved in Crohn’s disease (Sokol et al).
About triggering factors, P5: Authors should add diabetes to triggering factors as they have been reported by several authors (Garg, Bui and others) to be more prevalent in HS patients than controls and since diabetes is well-known to trigger diseases with an infectious component. Indeed, balancing diabetes in these patients really helps the disease.
About environmental factors, authors might also question a potential role of NSAIDs, since several studies report their common use in about 60-70% HS patients for pain (cf literature and SHSA 2019, communication). NSAIDs decrease immunity and have been reported to induce necrotizing fasciitis in patients treated for erysipelas with NSAIDs. Some authors have also reported their deleterious effect in HS patients, (Becherel, communication at EHSF, Athens, 2020; Nassif, SHSA, poster 2018; Nassif, SHSA, 2019, communication). Stopping NSAIDs can be an important environmental measure to take for patients’ improvement.
P7: another pillar should be maintenance treatment, in moderate and severe patients at least, since surgery only removes scars loaded with biofilms, but does not remove the immune deficiency, therefore does not prevent relapses in new areas.
A few typos:
- P2, line 64 experienced
- p2 line 82-83 it
- p6 line 221 players
- P8 line 333 based on the, 336: effectiveness
Author Response
Point to point reply:
We are very thankful to the Reviewer for the constructive suggestions, and we hope that the revised version of our manuscript can now reach high enough quality to be published in Cells.
Comment #1: This is a well-written review on HS field. A few precisions on microbiology, pathogenesis, triggering and environmental factors could improve this article. About microbiology :P5, line 204-208: When prolonged cultures (1 week) and/or bacterial metagenomics are performed, a rich anaerobes flora (including mostly porphyromonas and prevotella), but also actinomycetes and S milleri are isolated from HS lesions, varying with Hurley severity and, contrarily to what the authors wrote, there is very little S aureus in HS lesions, but instead rather S lugdunensis. The chosen references are not appropriate: the 4 main teams who extensively worked on microbiology should be cited: Guet-Revillet H, Coignard-Biehler H, Jais JP, Quesne G, Frapy E, Poirée S, Le Guern AS, Le Flèche-Matéos A, Hovnanian A, Consigny PH, Lortholary O, Nassif X, Nassif A, Join-Lambert O. Bacterial pathogens associated with hidradenitis suppurativa, France. Emerg Infect Dis. 2014 Dec;20(12):1990-8. doi: 10.3201/eid2012.140064; Guet-Revillet H, Jais JP, Ungeheuer MN, Coignard-Biehler H, Duchatelet S, Delage M, Lam T, Hovnanian A, Lortholary O, Nassif X, Nassif A, Join-Lambert O. The Microbiological Landscape of Anaerobic Infections in Hidradenitis Suppurativa: A Prospective Metagenomic Study. Clin Infect Dis. 2017 Jul 15;65(2):282-291. doi: 10.1093/cid/cix285; Ring HC, Bay L, Nilsson M, Kallenbach K, Miller IM, Saunte DM, Bjarnsholt T, Tolker-Nielsen T, Jemec GB. Bacterial biofilm in chronic lesions of hidradenitis suppurativa. Br J Dermatol. 2017 Apr;176(4):993-1000. doi: 10.1111/bjd.15007; Naik HB, Jo JH, Paul M, Kong HH. Skin Microbiota Perturbations Are Distinct and Disease Severity-Dependent in Hidradenitis Suppurativa. J Invest Dermatol. 2020 Apr;140(4):922-925.e3. doi: 10.1016/j.jid.2019.08.445; Schneider AM, Cook LC, Zhan X, Banerjee K, Cong Z, Imamura-Kawasawa Y, Gettle SL, Longenecker AL, Kirby JS, Nelson AM.J Loss of Skin Microbial Diversity and Alteration of Bacterial Metabolic Function in Hidradenitis Suppurativa. Invest Dermatol. 2020 Mar;140(3):716-720. doi: 10.1016/j.jid.2019.06.151. Reply: Many thanks for your comment. The part on microbiology has been completely changed and all the suggested references have been cited and discussed, substituting the previous ones.
Comment #2: P7, l 279: the cited ref 77 does not support the assumption of “low efficacy of AB”. In this reference, the authors state : “Ertapenem is highly effective but is reserved as third-line therapy for a single 6-week course as rescue therapy or during surgical planning.” Ertapenem happens to be an antibiotic strategy targeted according to Hurley 3 microbiology, demonstrating the correlation between efficacy and targeted antibiotherapy, instead of random antibiotics which do have a “low efficacy”. Reply: In the manuscript we did not state that there is “low efficacy” of systemic antibiotics. We only state that there is little evidence for efficacy of “topical antibiotics”, meaning there are not many publications on the efficacy of topical antibiotics in HS. The lack of evidence for topical antibiotics is supported by the statement in the cited reference [Alikhan et al]: “The only topical antibiotic studied is clindamycin 1% solution...”. We agree that there are a few promising systemic agents like ertapenem and also discuss tetracyclines, rifampicin and clindamycin as first-line agents. Ertapenem has been now added to the manuscript (paragraph 5.1).
Comment #3: About pathogenesis, authors could also mention AHR pathway (Prens et al) involvement as well as Tryptophane pathway (Guenin-Macé et al). Indeed, these 2 pathways interact and tryptophane pathway has also been involved in Crohn’s disease (Sokol et al). Reply: Many thanks for your comment. AHR pathway and Trypthophane pathway has been added in the pathogenesis section highlighting their potential role in linking the immunological and microbiological features of HS lesions
Comment #4: About triggering factors, P5: Authors should add diabetes to triggering factors as they have been reported by several authors (Garg, Bui and others) to be more prevalent in HS patients than controls and since diabetes is well-known to trigger diseases with an infectious component. Indeed, balancing diabetes in these patients really helps the disease. Reply: Thank you for pointing this out. As suggested, we added diabetes to triggering factors (paragraph 4.2). We have also added the suggested references.
Comment #5: About environmental factors, authors might also question a potential role of NSAIDs, since several studies report their common use in about 60-70% HS patients for pain (cf literature and SHSA 2019, communication). NSAIDs decrease immunity and have been reported to induce necrotizing fasciitis in patients treated for erysipelas with NSAIDs. Some authors have also reported their deleterious effect in HS patients, (Becherel, communication at EHSF, Athens, 2020; Nassif, SHSA, poster 2018; Nassif, SHSA, 2019, communication). Stopping NSAIDs can be an important environmental measure to take for patients’ improvement. Reply: Thanks for your valuable comment. We have now reported in the manuscript (point 4.2) your suggestions!
Comment #6: P7: another pillar should be maintenance treatment, in moderate and severe patients at least, since surgery only removes scars loaded with biofilms, but does not remove the immune deficiency, therefore does not prevent relapses in new areas. Reply: We agree that maintenance therapy is important. Maintenance therapy, though, includes mentioned “basic therapeutic measures” and, if necessary, anti-inflammatory therapy. We have added now a sentence pointing out the importance of basic therapeutic measures in order to maintain disease control. Moreover, we have now specified that “Combined with surgery, optimized antibiotic treatments may be promising in severe HS…..” (paragraph 5.1).
Comment #7:A few typos: P2, line 64 experienced; p2 line 82-83 it; p6 line 221 players; P8 line 333 based on the, 336: effectiveness. Reply: Thanks a lot!
Reviewer 3 Report
The introduction lacks some references to the altered quality of life of HS patients. There is plenty of data on this topic so you should not rely only on a previous review article. According to another classficiation by van der Zee (2015), there are even more than 3 phenotypes of HS - eg. ectopic subtype.
A good review article requires a color figure summarizing the crucial pathogenetic factors of the disease. This is as valuable as the extensive explanation in the text (which is very comprehensive and up to date). Concerning the treatment: "(...) surgical therapy of fistulas and destructed tissue". This explanation of surgical therapy's value sounds a bit strange. Is it only about fistulas? And what is "destructed tissue"?
I'm very sorry but I don't like the overuse of ref. 50 and 77 in the context of HS topical and systemic therapy. You chose the easiest way to cite the most important guidelines, thus omitting the cumbersome need to at least briefly mention the most important original publications on HS pharmacotherapy.
Yet please bear in mind - your article is not a meta-review. I agree that based on EBM standards the data on classic drugs in HS is usually not of the highest quality but your choice of words may imply that there are only minor case reports and genereally "no need to talk about it because its useless anyway". There are nine authors of this manuscript who are experts - so do everything the expert way.
To preserve the reasonable balance between a narrative review component (your basic concept of this manuscript) and a systematic review (resulting in more comprehensive impression of the manuscript) a simple table summarizing the most crucial topical and systemic drugs can be introduced, grouped according to their mechanisms and supported by adequate references. Nothing much - but so important!
Author Response
Point to point reply:
We are very thankful to the Reviewer for the constructive suggestions, and we hope that the revised version of our manuscript can now reach high enough quality to be published in Cells.
Comment #1: The introduction lacks some references to the altered quality of life of HS patients. There is plenty of data on this topic so you should not rely only on a previous review article. Reply: Thank you very much for your comment. We have now added more references.
Comment #2: According to another classficiation by van der Zee (2015), there are even more than 3 phenotypes of HS - eg. ectopic subtype. Reply: Thanks a lot for your comment, all the subtypes have been added to the manuscript.
Comment#3: A good review article requires a color figure summarizing the crucial pathogenetic factors of the disease. This is as valuable as the extensive explanation in the text (which is very comprehensive and up to date). Reply: We have now integrated “the figure 3” reporting a schematic overview of HS pathogenesis.
Comment#4: Concerning the treatment: "(...) surgical therapy of fistulas and destructed tissue". This explanation of surgical therapy's value sounds a bit strange. Is it only about fistulas? And what is "destructed tissue"? Reply: Thank you for pointing this out. We changed the wording and clarified what are the major targets of surgical intervention in paragraph 5.1.
Comment #5: I'm very sorry but I don't like the overuse of ref. 50 and 77 in the context of HS topical and systemic therapy. You chose the easiest way to cite the most important guidelines, thus omitting the cumbersome need to at least briefly mention the most important original publications on HS pharmacotherapy. Reply: We have now added original publications on HS pharmacotherapy.
Comment#6: Yet please bear in mind - your article is not a meta-review. I agree that based on EBM standards the data on classic drugs in HS is usually not of the highest quality but your choice of words may imply that there are only minor case reports and genereally "no need to talk about it because its useless anyway". There are nine authors of this manuscript who are experts - so do everything the expert way. Reply: As stated in point 5.1, we value the advances that have been made in pharmacotherapy of HS in the past decades. Nevertheless, from own experience and published data, we want to emphasize the necessity of surgical interventions in advanced disease stages where sinus tracts, large nodules, and scars are present. We changed the wording of the last sentence of point 5.1 in order not to give the impression that there is only surgery as a therapy option in HS.
Comment#7: To preserve the reasonable balance between a narrative review component (your basic concept of this manuscript) and a systematic review (resulting in more comprehensive impression of the manuscript) a simple table summarizing the most crucial topical and systemic drugs can be introduced, grouped according to their mechanisms and supported by adequate references. Nothing much - but so important! Reply: Thank you for the constructive feedback. We included a table (Table 1) showing the therapeutic options in HS with adequate references.
Round 2
Reviewer 3 Report
The authors have adequately answered all the comments